# Hiding in a Plain Sight: Out-of-Distribution Detection from Logit Space Embeddings

## Abstract

Although deep learning (DL) models have revolutionized the field of machine learning (ML), these classification models cannot easily distinguish the in-distribution (ID) versus the out-of-distribution (OOD) data at the test phase. This paper analyzes the landscape of ID and OOD data embeddings and demonstrates that OOD data is always embedded toward the center in the logit space. Furthermore, IDs data are embedded far from the center towards the positive regions of the logit space, thus ensuring minimal overlap between ID and OOD embeddings. Based on these observations, we propose to make the classification model sensitive to the OOD data by incorporating the configuration of the logit space into the predictive response. Hence, we estimate the distribution of the ID logits by utilizing a density estimator over the training data logits. Our proposed approach is data and architecture-agnostic and could be easily incorporated with a trained model without exposure to OOD data. We ran experiments on the popular image datasets and obtained state-of-the-art performance and an improvement of up to 10% on AUCROC on the Google genome dataset.

## 1 Introduction

Deep learning (DL) classification models can generalize over the discriminative features of a large amount of data, thus providing higher classification accuracy than alternative models. The predictive response of DL models is highly accurate whenever the test data falls within the training data distribution. However, these models fail for out-of-distribution (OOD) data, as they operate under the strong assumption that the test item belongs to one of the designated classes. This incapability of DL classifiers limits their adaptation into sensitive application areas such as biomedicine. E.g., when classifying bacteria from genome sequences using a DL model, it is crucial to consider the presence of novel (i.e., OOD) bacteria. Failing to account for them may result in the incorrect classification of these novel bacteria as one of the already known types (Ren et al., 2019).

To tell OODs apart from IDs, today's deep learning (DL) architectures try to estimate the statistical uncertainty over the discriminative features of the training data (Kirichenko et al., 2020). However, all the previous methods implicitly assume random scattering of the OOD relative to the ID in the embedding space and fail to provide an easy-to-use solution for OOD detection. Instead, we demonstrate that a well-trained DL classifier with nonlinearities that suppresses negative values (e.g., ReLU) projects the ID data into class-wise clusters toward positive regions and far from the center of the logit space (cf., fig. 1a). Furthermore, we show (analytically and empirically) that OOD data are not arbitrarily scattered in the logit space but hidden in plain sight at its center (cf., fig. 1c). These low-magnitude logit values for OODs result directly from their statistical independence relative to the trained model's parameter. Hence, ensuring minimal overlap between OODs and IDs. *Although previous works have identified and experimented with the separation of OODs and IDs in the logit space (Lee et al., 2018; Liu et al., 2020), to the best of our knowledge, this is the first work that demonstrates the expected configuration of OODs and IDs.* As a result of this identified separation, we can safely construct an accurate ID detector with simple architecture (cf., fig. 1c). In practice, targetable OODs as training data are necessary to depict the regions in the logit space where the OODs are projected, thereby enabling their detection. Since the distribution of OODs is unbounded, consolidating a proper training set composed solely of targetable OODs is not feasible. Thus, we cast the problem as

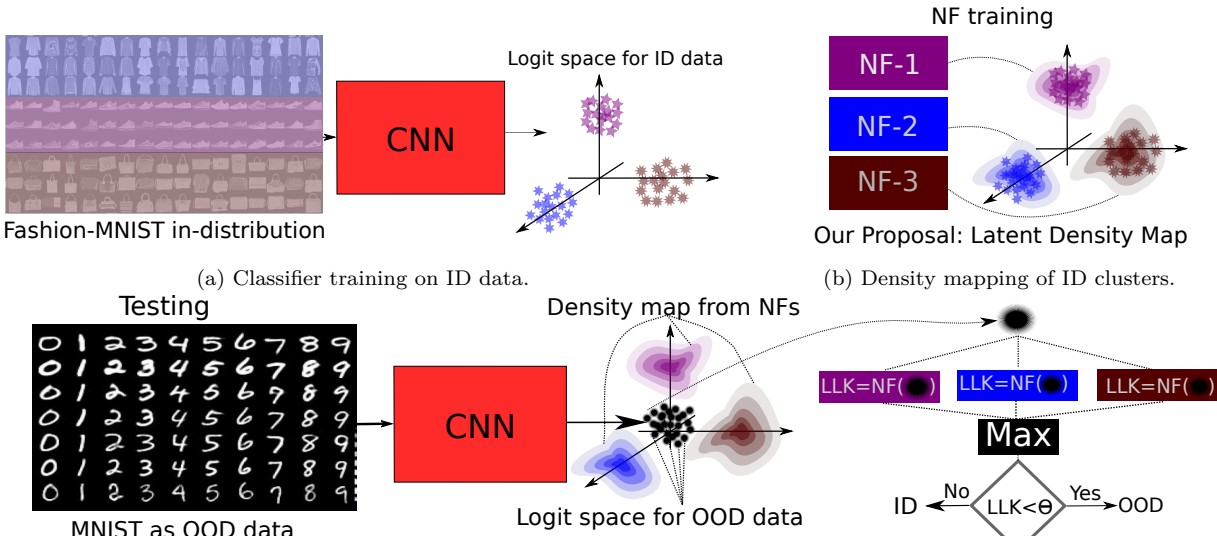

(a) Classifier training on ID data.                (b) Density mapping of ID clusters.

(c) During the test phase, the OOD uses CNN to produce logit embeddings. If the likelihood (LLK) of these embedding is lower than a threshold ($\theta$), it is classified as OOD; otherwise, it is ID.

Figure 1: A CNN classifier model that has undergone training projects the data towards positive regions into clusters specific to each class fig. 1a. An individual density estimator is applied to each class-wise cluster formed by the logit projections fig. 1b. The identification of OOD data is achieved by analyzing the likelihood of embeddings in the logit space fig. 1c.

ID detection, where any non-ID is safely considered OOD. We represent the densities of each ID cluster to estimate the likelihood of any data being ID. Moreover, any data that attains a likelihood value below a certain threshold should fall outside any ID cluster and be considered OOD (cf., fig. 1c).

We identified normalizing flows (NF) as a good candidate for density estimation since it provides exact likelihood without altering the dimensionality of the data Papamakarios et al. (2021). Since OODs and IDs logits are separated, the proposed method admits a simple NF architecture for each ID cluster, circumventing the need for exposure to real (or synthetic) OOD data, and does not demand alternation to the topology of the DL classifiers. The contributions of the paper are the following:
1) Analytical and empirical evidence for the ID and OOD data positioning in the logit space;
2) Novel highly effective framework for OOD detection using density estimation over the logits;
Despite having a reduced model complexity, the proposed approach (cf., fig. 1b) matches state-of-the-art (SOTA) models' performance in grayscale and colored images. Furthermore, our experiments show that it considerably improves the OOD detection performance relative to the previously reported baselines on the Google genome dataset.

## 2 Method

### 2.1 In-distribution data positioning in the logit space.

Training a deep learning (DL) classifier involves utilizing the cross-entropy loss, denoted as $H(Y, \hat{Y}) = -\sum_i Y(i) \log(\hat{Y}(i))$, to encourage the prediction ($\hat{Y}$) to closely align with the ground truth (Y). When employing one-hot encoding for both the prediction ($\hat{Y}$) and ground truth (Y), the training objective simplifies to

$$H(Y, \hat{Y}) = -\sum_i Y(i) \log(\hat{Y}(i)) = \underbrace{-Y(j) \log(\hat{Y}(j))}_{Y(j)=1, j \to \text{correct class}} - \sum_{i, i \neq j} \underbrace{Y(i) \log(\hat{Y}(i))}_{Y(i)=0, i \to \text{incorrect class}} = -\log(\hat{Y}(j)). \quad (1)$$

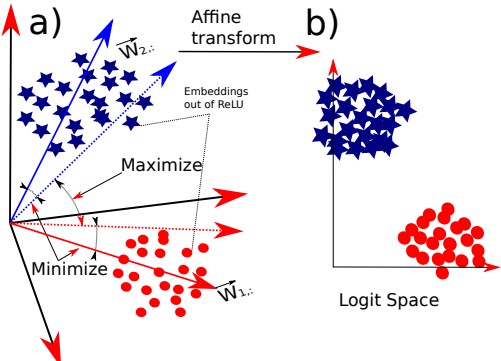

Figure 2: This toy example shows the separation of ID in a binary classification task. Figure a) contains the embeddings ($E$) rectified with a ReLU. Figure b) shows the linear separation of class-wise clustering of ID data logits ($\hat{L}$). The smaller the angle between $\vec{E}$ and $\vec{W}_{1,:}$, the higher the dot-product $\langle W_{1,i}, E_i \rangle$ Figure a); thus the more distanced from the center the ID logits are (Figure b). The bigger the angle between ($\vec{E}$) and $\vec{W}_{2,:}$, the higher the dot-product $\langle \vec{W}_{2,i}, \vec{E}_i \rangle$ (cf., fig. 2 a), the more compact the ID logits are.

Eventually, the minimization cross-entropy loss $\big(i.e., \min[H(Y, \hat{Y})]\big)$ equivalues to maximum likelihood estimation (MLE) $\big(i.e., \min[-\log(\hat{Y}(j))]\big)$. As training progresses, the softmax layer aims to generate a response close to one for the cell corresponding to the correct class $\big(i.e., \hat{Y}(j) \to 1\big)$. Additionally, due to the property that the softmax output is confined within a simplex $\big(i.e., \hat{Y}(j)^{\uparrow} + \sum_{i,i \neq j} \hat{Y}(i)^{\downarrow} = 1\big)$, the remaining cells are pushed towards values close to zero $\big(i.e., \hat{Y}(i)_{i \neq j} \to 0\big)$.

Hence, optimization in this context can be seen as maximizing the softmax cell for the correct class while minimizing the cells for the incorrect classes. This optimization applies directly to the corresponding logit cells since softmax keeps the order of logits intact.

Specifically, the logit cell (i.e., $\hat{L}(j)$) associated with the correct class aims to achieve large positive values, while the logit cells (i.e., $\hat{L}(i)_{i \neq j}$) for the incorrect classes aim for small values. However, whenever ReLU is used as an activation layer, we demonstrate that the minimization process results in logit values near zero rather than small negative magnitudes.

**Theorem 1** *When training a DL classifier with ReLU (Rectified Linear Unit) as the nonlinear activation function, the logit associated with the correct class endeavors to achieve high magnitudes of positive values $\hat{L}(j) \to +\infty$. Simultaneously, the remaining cells representing incorrect classes aim to attain low-magnitude values $\hat{L}(i)_{i \neq j} \to 0$.*

To prove this theorem, it is necessary to state the following lemma:

**Lemma 1** *In the positive region of high-dimensional space, the maximum angle two vectors can achieve is perpendicular (cf., proof in Appendix A).*

To prove the restriction towards zero of the logit cells not corresponding to the correct class ($\hat{L}(i)_{i \neq j} \to 0$), it is paramount to note that the predecessor latent space ($\hat{E}(i)$) is restricted towards the positive values due to the ReLU (cf., fig. 2.a). The layer preceding the softmax is a linear transformation of the data from high-dimensional embeddings ($\hat{E}$) to the logit space ($\hat{L} = \hat{E} \times W$, s.t: $\times$ is the matrix multiplication) with dimensions matching the number of designed classes(cf., fig. 2.b). Since the optimizer tries to attain maximum response for the logit cell $\hat{L}[i]$, it should maximize the dot-product $\arg\max_{W[i,:]} \langle \hat{E}[:], W[i,:] \rangle^1$,s.t: $\hat{E}[:] \geq 0$.

Considering the embeddings $\hat{E}[:]$ and $W[i,:]$ as a vector in the vector space (cf., fig. 2 a), $\langle \vec{E}[:], \vec{W}[i,:] \rangle$ maximization results in angle minimization between $\vec{E}[:]$ and $\vec{W}[i,:]$ $\big(i.e., \min \angle(\vec{W}[i,:], \vec{E}[:])\big)$ while the

---

[1] $\langle , \rangle$ indicates the dot-product

former always remain in the positive regions. The optimization tries to keep the direction of the vector $\vec{W}[i, :]$ similar to the cluster of vectors $\vec{E}[:]$, namely in the positive regions (cf., fig. 2.a).

Furthermore, the optimization tries to attain a minimum response for every other logit cell $\hat{L}[j \neq i]$ that does not correspond to the correct class as $\arg\min_{W[j \neq i, :]} \langle \hat{E}[:], W[j \neq i, :] \rangle$, s.t: $\hat{E}[:] \geq 0$. Namely, maximizing the angle between $\vec{W}[j \neq i, :]$ and the cluster of vector data $\vec{E}[:]$, $\left(i.e., \max \angle(\vec{W}[j \neq i, :], \vec{E}[:])\right)$ (cf., fig. 2.a).

Hence, the clusters belonging to different classes strive to achieve maximum angular separation from one another, and the parameter vectors $\vec{W}[i, :]$ align accordingly. As all vectors $\vec{E}[:]$ are angularly separated within the positive region, the maximum angle between these two vectors is close to perpendicularity (cf., Lemma 1). Therefore, the minimized logit values $\left(\arg\min(\vec{W}[j \neq i, :], \vec{E}[:]) \approx 0\right)$ would approach assymptotically to zero during training.

Consequently, the asymptotic behavior of the data configuration in the logit space compels the data points to form compact clusters far from the center of the space, corresponding to their respective classes. This process leads to the minimization of interclass distances and the maximization of intraclass distances.

## 2.2  Out-of-distribution data positioning in the logit space

We demonstrated that optimization pushes the IDs away from the center of the logit space and toward the positive regions (cf., Section 2.1). To show that OODs are not arbitrarily scattered in the logit space but projected towards the center, we show that the interaction of the parameters of the model ($\omega$) and the OOD data ($x_{OOD}$) is upper-bounded by the independence consistency (cf., Definition 1) between these two random variables (r.v) ($\omega \perp x_{OOD}$)[2]. The necessity for this definition primarily arises from the limitless probability space of the data (x), in contrast to the static nature of the probability distribution of the weights ($\omega$) once the network completes its training. Namely, the new estimate $\hat{cov}(x, \omega)$ is inherently a random value, whereas its expected value and variability are deterministic values.

**Definition 1** *(Consistently independent) Given two r.v $x, \omega \in R$ where the probability density function (pdf) of $x$ is fixed, whereas the pdf of $\omega$ is unbounded. These two r.v are systematically independent if the expectation of their empirical covariance is zero (i.e., $E[\hat{cov}(x, \omega)]^2 = 0$). Additionally, these two r.v maintain this independence consistently whenever the variability of their empirical covariance is zero (i.e., $Var[\hat{cov}(x, \omega)] = 0$). Hence two r.v are consistently independent if and only if $E[\hat{cov}(x, \omega)]^2 + Var[\hat{cov}(x, \omega)] = 0$.*

Empirical covariance $\left(i.e., \hat{cov}(x, \omega) = \frac{1}{(n-1)} \sum_{i=1}^{n}(x_i - \mu_x)(\omega_i - \mu_\omega)\right)$ is a numerical assessment of the covariance. A high magnitude of the empirical covariance characterizes two variables that covary together in their respective spaces and vice versa.

To establish the bounds of OOD in the logit space, we start by showing that a dot-product between two variables acts as a lower bound for their empirical covariance value (Corollary 1).

**Corollary 1** *Given two r.v $(x, \omega \in R)$, the magnitude of their dot-product is a lower bound for the magnitude of their empirical covariance (cf., eq. (2)) (cf., proof in Appendix B).*

$$\sum_{i=1}^{n} x_i \omega_i \leq (n-1)\left|\hat{cov}(x, \omega)\right|, \forall x, \omega \in \mathbb{R} \tag{2}$$

Thus, the empirical covariance between these two entities (i.e., weights ($\omega$) and training data $x_{\text{ID}}$) are maximized while their dot-product magnitude is maximized during the training process. While the $\left|\hat{cov}(x, \omega)\right|$ is maximized, the $\left|\hat{cov}(x_{\text{OOD}}, x_{\text{ID}})\right|$ is assumed to be minimal since ($x_{\text{OOD}} \perp x_{\text{ID}}$)) the following corollary can be easily derived (Corollary 2).

---

[2]$\perp$ for two r.v means statistical independence.

**Corollary 2 *(Co-variability)*** *Since the OOD ($x_{OOD}$) and ID ($x_{ID}$) data come from two different distributions altogether, one can assume that their empirical covariance is minimal (i.e., $côv(x_{OOD}, x_{ID}) \approx 0$). Given that IDs covary with $\omega$ but not with the OODs, we prove that OODs do not covary with the $\omega$ (i.e., $côv(x_{OOD}, \omega) \approx 0$) (cf., proof in Appendix C).*

The training process makes the model parameters ($\omega$) more statistically dependent on the IDs ($x_{\text{ID}}$), while the latter are supposed to be consistently independent of the OODs ($x_{\text{OOD}}$). Since the magnitude of the logits is a result of dot-product, this can be upper-bounded by the independence consistency between the OODs and $\omega$. Utilizing this low empirical covariance of OODs ($x_{\text{OOD}}$) and the model weights ($\omega$), we derived an upper bound for the expectation of their dot-product (cf., Theorem 2).

**Theorem 2 (Expectations of OOD embeddings)** *Given two random variables ($x_{OOD}, \omega$) the expectation of their dot-product is upper bounded by their independence consistency (cf., proof in Appendix D):*

$$\left| E\left\{ \sum_{i=1}^{n} x_{OOD_i} * \omega_i \right\} \right| \leq (n-1) \left| \sqrt{E\{côv(x_{OOD}, \omega)\}^2 + var\{côv(x_{OOD}, \omega)\}} \right|, \forall x_{OOD}, \omega \in R \qquad (3)$$

The r.h.s of eq. (3) indicates the OODs expected distance from the center of the logit space. The l.h.s eq. (3) is the independence consistency between OOD data and the parameters of the models. Therefore, the better the trained model is (i.e., $\left| \sum_{i=1}^{n} x_{\text{OOD}_i} * \omega_i \right|^{\uparrow}$ is high), the farther from the center the IDs are clustered (cf., Section 2.1). The more different OODs are from the IDs, the lower their covariance with the IDs, hence the lower $\left| côv(x_{\text{OOD}}, \omega) \right|^{\downarrow}$, leading to lower l.h.s of eq. (3). Hence, unlike ID data, the OOD data embeddings will not be able to produce high magnitude values for any of the logit cells, and their embeddings are squeezed more towards the center and well separated from IDs.

To restrict the visualization to 3D space, a three-class model (Resnet-34) is investigated on the CIFAR-3[3] vs. SVHN scenario. The untrained Resnet-34 model with three output classes whose weights are randomly initialized projects both CIFAR-3 and SVHN datasets towards the center of the logit space (cf., fig. 3).

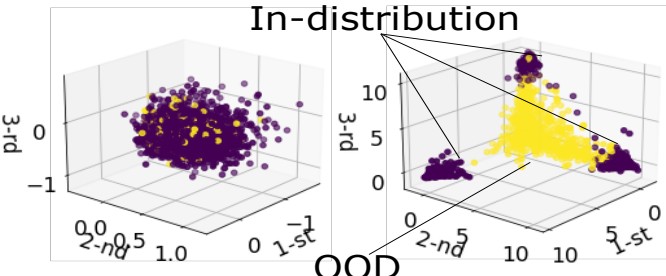

Figure 3: CIFAR-3 as ID and SVHN as OOD. *Left*: Before the training, both ID and OOD maintain the tendency towards the center of the latent space. *Right*: After the training, the ID data are clustered, whereas the OOD persists towards the center.

Our observation for a trained model shows that ID training data attains high positive values for the logit cells corresponding to the correct class based on the kernel density estimation (KDE) in fig. 4a and fig. 6a in Appendix. Simultaneously, all the other logit corresponding to the incorrect cell maintain values at the proximity of zero (cf., fig. 4c and fig. 6b in Appendix). As expected, the OOD logits tend towards the proximity of zeros for all cells (cf., fig. 4e and fig. 6c in Appendix). This concentration of OODs logit the center is further validated empirically for both images (grayscale and colored) as well as genome dataset (cf., figs. 7 to 11 in Appendix).

Moreover, diverse activation functions (Relu, Leaky Relu, Celu (Barron, 2017), Gelu (Hendrycks & Gimpel, 2023), Selu (Klambauer et al., 2017), Elu (Clevert et al., 2016), Silu (Elfwing et al., 2017), Mish (Misra,

---

[3]Only three classes from CIFAR-10

2020)) have been tested to understand the significance of constraining negative values. One can notice that Relu and Leaky Relu have the best separation between OODs and IDs and the smallest spread in the logit space (cf., figs. 4b, 4d and 4f and figs. 14 and 15 in the Appendix for individual plots for each activation). Meanwhile, the rest of the activations have either a higher spread or a smaller displacement between OODs and IDs in the logit space (cf., figs. 4b, 4d and 4f and figs. 13 and 16 to 21 in the Appendix for individual plots of each activation).

Another important factor is the shape of the ID data does not follow a normal distribution necessarily (cf., fig. 4a), especially when the network is small (cf., fig. 8 in the Appendix). This is an important aspect of the proposed method that can account for any shape for the ID logits, unlike Lee et al. (2018), which assumes a strict Gaussian distribution.

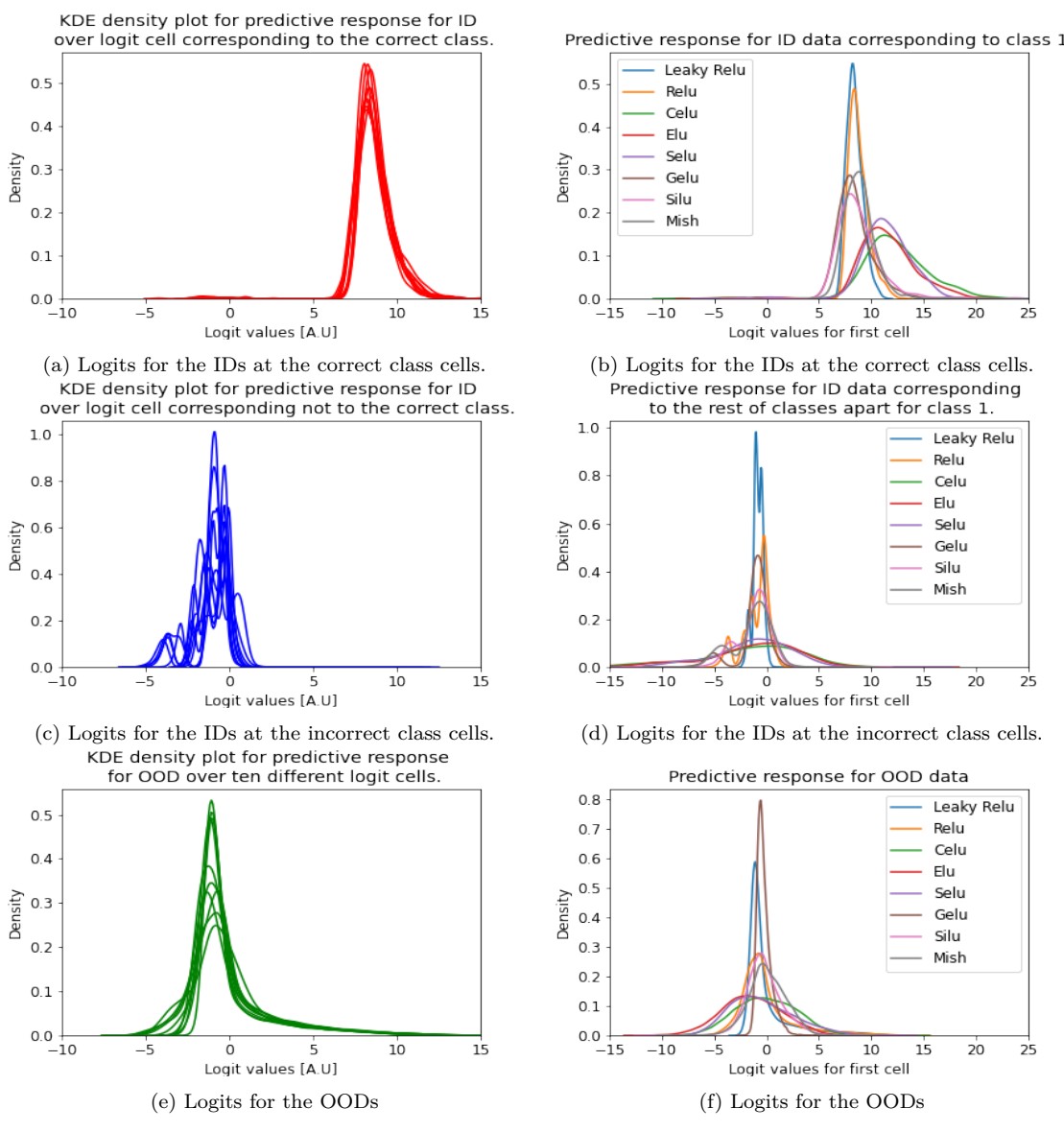

(a) Logits for the IDs at the correct class cells.

(b) Logits for the IDs at the correct class cells.

(c) Logits for the IDs at the incorrect class cells.

(d) Logits for the IDs at the incorrect class cells.

(e) Logits for the OODs

(f) Logits for the OODs

Figure 4: KDE response SVHN (ID) vs CIFAR-10 (OOD) while using Resnet-34 with ReLU activation function.

### 2.3 Model for OOD detection

After establishing two distinct regions for OOD and ID data that do not overlap, the next step involves detecting and exposing OODs during the testing phase.

In practice, accurately defining the boundaries of the OOD region is challenging because OOD data is typically unavailable. However, even after training the classifier, we retain access to the embeddings of the ID training data. Leveraging this information, it becomes feasible to delineate the ID regions using density estimation techniques.

Therefore, we address the issue of OOD absence by developing an ID detection system that acts as a one-class classifier, treating any data identified as non-ID to be OOD. To delineate the regions corresponding to ID, we utilize the density representation of each ID cluster to assess the probability of a given data point being classified as ID. By creating a density map based on the occurrence of ID embeddings from the training data, we capture the uncertainty associated with the discriminative features.

Normalizing flows (NF) offers an appropriate framework for density estimation due to their ability to estimate the likelihood for a given data while preserving the original dimensionality of the vector space (Papamakarios et al., 2021). While alternative parametric density estimators like the Gaussian distribution provide similar capabilities, they rely on a more restrictive assumption that the data distribution follows an elliptic shape. In contrast, NF does not impose any prior assumptions about the shape of the target distribution. Instead, they employ a bijective transformation, denoted as $u = T_\theta^{-1}(x)$, to map a simple base distribution $P_U(u)$ which could be Gaussian to match the desired target distribution $P_X(x)$ (cf., eq. (4)).

$$P_X(x) = P_U(T_\theta^{-1}(x)) \left| det\{J_{T_\theta^{-1}}(T_\theta^{-1}(x))\} \right| \tag{4}$$

The density estimation augments multiple runs of individual data on the classifier model as NF extrapolates the density values in the continuity of the latent space. NF can interpolate a likelihood value from positioning a single data in the latent space, reflecting the frequency of past occurrences.

The empirical distribution of the logits is already known to have the same modality as the number of clusters. Under this domain knowledge, it is more beneficial to employ multiple NF for each class-wise ID cluster with simple architecture instead of a single but complex NF (cf., algorithm 2). Utilizing NF for each ID cluster (cf., algorithm 2 and fig. 22 in Appendix), it is possible to maintain a high likelihood at high-density regions and, by default, a low likelihood elsewhere, including the OOD region (cf., algorithm 2 and Appendix I). Then, any data whose likelihood is below a certain threshold are considered OOD (cf., fig. 1 and algorithm 1).

---

**Algorithm 1** OOD detection
___

**Input:** Trained classifier that produces logits $\hat{L}(x) = F_\theta(x)$. Individual data $x$. The number of classes K. K different trained $NF_{[1 \cdot K]}$ models. Threshold value $\theta$.

1: **procedure** D(x)
2:      $\hat{L}(x) \leftarrow F_\theta(x)$                  ▷ Get the embeddings in the logit space
3:      $LogLikehood_{Max} \leftarrow 0$            ▷ Get the embeddings in the logit space
4:      **for** each $c = 1, 2, \ldots K$ **do**             ▷ Iterate each $NF_{[1 \cdot K]}$
5:          $\hat{LLK} \leftarrow NF_{[c]}(L(x))$           ▷ Get the likelihood for class c
6:          **if** $\hat{LLK} = LogLikehood_{Max}$ **then**     ▷ If the prediction matches the given label
7:             $LogLikehood_{Max} = \hat{LLK}$     ▷ Train the NF over the ID cluster of class $c$
8:      **if** $LogLikehood_{Max} \geq \theta$ **then**          ▷ Check if $x$ is an ID
9:          $x \rightarrow ID$
10:     **else**
11:         $x \rightarrow OOD$              ▷ If the data is not ID, then it is OOD

---

---

**Algorithm 2** Training of NF for each ID cluster

---

**Input:** Trained classifier that produces logits $\hat{L}(x) = F_\theta(x)$. ID annotated training data $[c, x] \in X$. The number of classes K. K different $NF_{[1 \cdot K]}$ models.

1: **procedure** TRAIN INDIVIDAL NF
2:     **for** each round $t = 1, 2, \ldots$ **do**
3:         **for** each data $c, x \in X$ **do**                ▷ Iterate the dataset in batches
4:            $\hat{L}(x) \leftarrow F_\theta(x)$           ▷ Get the embeddings in the logit space
5:            $\hat{c} \leftarrow \arg\max L(x)$         ▷ Get the class prediction
6:            **if** $\hat{c} = c$ **then**         ▷ If the prediction matches the given label
7:                Train $\rightarrow NF_{[c]}[L(x)]$     ▷ Train the c-th NF over the ID cluster of class $c$

---

## 3  Related works

**Theoretical studies:** In a recent exploration of the learnability of the OOD task using the lens of probably approximately correct (PAC) theory, researchers conducted an insightful analysis (Fang et al., 2022). Furthermore, a recent empirical investigation focused on the transferability of ID training to OOD detection (Wenzel et al., 2022). An important finding from this research was the asserted correlation between enhanced ID training and improved OOD detection performance. Additionally, examination of the OOD region within the softmax space has been explored in related works (Pearce et al., 2021; Frosst et al., 2019).

**Classification-based:** Detecting OOD samples using a classifier trained on ID data relies on prediction scores that are used to tell the ID classes apart. Early works use probability values from softmax as a common choice. The baseline approach for OOD detection involved using the maximum softmax output as a guiding principle . Other methods have been explored to estimate uncertainty in predictive responses by creating an ensemble of models (Vyas et al., 2018). Nevertheless, this ensemble-based approach requires training and storing multiple models.

Alternatively, some methods aim to estimate data and model uncertainty using an ensemble of models. The generalized uncertainty is then distilled into a single model, and OOD samples are detected using the distilled uncertainty from this single model instead of an ensemble (Vadera et al., 2020a; Malinin et al., 2019; Vadera et al., 2020b; Depeweg et al., 2017; Lakshminarayanan et al., 2016).

ODIN is another method that enhances sensitivity towards OOD data by maximizing the entropy of softmax responses (Liang et al., 2020). ODIN increases OOD detection capabilities by combining a calibrated softmax with input perturbation. However, ODIN requires exposure to OOD data for training.

Another method based on softmax output proposes a regret score, calculated as the logarithm of the sum of fine-tuned probability values obtained from softmax (p(y|x)). However, softmax itself cannot capture sufficient uncertainty, making its application in OOD detection sub-optimal (Gal & Ghahramani, 2016; Hendrycks & Gimpel, 2016; Liu et al., 2020; Sun et al., 2021; Hendrycks et al., 2019; Sastry & Oore, 2020; Yu & Aizawa, 2019; Hein et al., 2019).

Another model utilizes the Mahalanobis distance (MD) between the test data and the per-class center in the latent space (Lee et al., 2018). While this method performs well on popular image datasets, it can fail on complex datasets where the logits do not follow a Gaussian distribution. Nonlinear boundaries in the embedding space must be considered, as the MD accounts only for isocontours of elliptic shape.

Energy-based models have also shown potential for OOD detection (Liu et al., 2020), although training them can be challenging. Another approach involves using Gram matrices of different orders from each layer's output (Sastry & Oore, 2020). However, its performance depends on the specific order chosen for the Gram matrix of each layer.

**Generative models:** Deep generative models have been extensively studied for out-of-distribution (OOD) detection due to their ability to represent high-dimensional data uncertainty in a parametric form (Serrà et al., 2020; Xiao et al., 2020; Wang et al., 2020; Choi et al., 2019; Kim et al., 2021; Schirrmeister et al.,

2020; Abati et al., 2019; Ren et al., 2019; Nalisnick et al., 2018). An additional advantage of these models is their ability to operate without relying on labels, which can often be challenging to obtain.

However, these models do not generalize over the discriminative features since their training process does not involve the context of the training data. Consequently, when trained on image data, these models tend to generalize based on pixel correlation values alone, without considering discriminative features. Hence, these models may assign likelihood values to OOD data that are similar to or even higher than those assigned to ID data (Kirichenko et al., 2020; Nalisnick et al., 2018; Kim et al., 2021; Xiao et al., 2020; Liu et al., 2020; Choi et al., 2019; Ren et al., 2019; Schirrmeister et al., 2020; Wang et al., 2020; Hendrycks et al., 2019; Serrà et al., 2020; Hsu et al., 2020; Sun et al., 2021; Sastry & Oore, 2020).

Instead, some recent work tries to leverage contrastive learning for feature distillation and then train a density estimator (Liu & Abbeel, 2020).

An improved alternative model estimates the ratio of the training data likelihood over the likelihood of their noisy version (Ren et al., 2019). The authors argue that performing density estimation over a noisy dataset equals uncertainty estimation over non-discriminative features. By incorporating uncertainty in the denominator, this method aims to reduce the likelihood assigned to background noise, thereby amplifying the core features. The effectiveness of this approach relies on the level of noise introduced during the training data generation process. Consequently, determining the appropriate noise level requires exposure to OOD data through simulation or real-world acquisition.

In contrast to previous approaches, our proposed solution anticipates the arrangement of both in-distribution (ID) and out-of-distribution (OOD) data within the logit space. This eliminates the necessity for complex models to achieve high performance in OOD detection.

## 4 Experiments

In order to evaluate the effectiveness of the suggested approach, the ID and OOD datasets must exhibit a significant degree of similarity while being semantically different. We performed experiments using diverse image datasets, encompassing grayscale images, colored images, and a genome dataset.

### 4.1 Performance evaluation

The proposed approach outperforms baselines on the genome dataset and grayscale images and performs on par or better on colored images.

The DL classifier model is trained to classify the ID training data correctly and, by extension, to obtain their logit space representation (cf., Appendix F, G and H). All the non-linearities in our method are ReLU to ensure maximum displacement from the center of the logit space of ID embeddings. In the case of the grayscale images, a small DL model with three convolutional layers and two fully connected layers (cf., 6 in Appendix and Appendix F.1) is utilized, while for the genome experiment, we utilize a Resenet-34 (He et al., 2015) (cf. Appendix H). In the experiments with colored images, a Resnet-34 and Densenet-121 (Huang et al., 2018b) are trained as a DL classifier model (cf., Appendix G).

Thereafter, the softmax layer of the classifier model was removed, and the remaining parameters were frozen, meaning their gradients were set to zero. The trained classifier then operates as a mapping function to convert the training data into logits. The logits of the ID training data are used to train individual NF with identical architectures. Each NF is dedicated to one of the classes (cf., algorithm 2, 1b in Appendix and Appendix I). Real-valued non-volume preserving (RealNVP) is the NF model choice (Dinh et al., 2016), which consists of multiple MLP layers (cf., table 12 in Appendix). The base distribution ($i.e., P_X(x)$) for the NF is a multivariate standard Gaussian with the masking as in I in the Appendix.

The performance of the method in detecting out-of-distribution (OOD) data was evaluated using the receive operating characteristic (ROC) and precision-recall curve (PRC) (cf., table 1) as well as true negative rate (TNR) at 95% true positive rate (TPR) (cf., table 2).

Table 1: The proposed method performs better on grayscale images and the genome dataset. Performance comparison on the Fashion-MNIST (ID) vs. MNIST (OOD) on the first part and genome dataset on the second. The mean performance is reported together with the variance of ten rounds in the brackets. Apart from our method result, the rest of the results are from Ren et al. (2019) and Fort et al. (2021)

| Dataset → | Fashion-MNIST vs MNIST | | Google Genome | |
|---|---|---|---|---|
| Methods ↓ | AUCROC [(%)↑] | AUCPRC [(%)↑] | AUCROC [(%)↑] | AUCPRC [(%)↑] |
| **Our Method (Ours)** | **99.2 (0.1)** | **99.4 (0.1)** | **84.1 (0.9)** | **85.9 (0.8)** |
| Likelihood Ratio ($\mu$) | 97.3 (3.1) | 95.1 (6.3) | 73.2 (1.5) | 71.9 (1.7) |
| Likelihood Ratio ($\mu, \lambda$) | **99.4 (0.1)** | **99.3 (0.2**) | 75.5 (0.1) | 71.9 (0.6) |
| Mahalanobis distance | 94.2 (1.7) | 92.8 (2.1) | 52.5 (1.0) | 50.3 (0.7) |
| $p(\hat{y}\|x)$ with calibration | 90.4 (2.3) | 89.5 (2.3) | 66.9 (0.5) | 63.5 (0.4) |
| ODIN | 75.2 (6.9) | 76.3 (6.2) | 69.7 (1.0) | 67.1 (1.2) |
| Ensemble, 5 classifiers | 83.9 (1.0) | 83.3 (0.9) | 68.2 (0.2) | 64.7 (0.2) |
| Ensemble, 10 classifiers | 85.1 (0.7) | 84.4 (0.6) | 69.0 (0.1) | 65.5 (0.2) |
| Ensemble, 20 classifiers | 85.7 (0.5) | 84.9 (0.4)) | 69.5 (0.1) | 65.9 (0.1) |
| BERT+ Mahalanobis | - | - | 77.5(0.04) | 78.8(0.06) |

Table 2: The proposed method performs comparably with the baselines on the colored image dataset. Performance comparison on the CIFAR-10, CIFAR-100, and SVHN while using Resnet and Densenet as a classifier. Apart from our method result, the rest of the results are from Sastry & Oore (2020).

| ID (model) | OOD | Baseline/Odin/Mahalanobis/Gram/Ours | |
|---|---|---|---|
| | | TNR at TPR 95% [(%)↑] | AUCROC [(%)↑] |
| CIFAR-100 (ResNet) | iSUN | 16.9 / 45.2 / 89.9 / 94.8 / **96.2** | 75.8 / 85.5 / 97.9 / 98.8 / **99.1** |
| | LSUN(C) | 18.7 / 44.1 / 64.8 / 64.8 / 69.1 | 75.5 / 82.7 / 92.0 / 92.1 / **95.1** |
| | TinyImgNet(C) | 24.3 / 44.3 / 80.9 / 88.5 / **91.8** | 79.7 / 85.4 / 96.3 / 97.7 / **98.5** |
| | SVHN | 20.3 / 62.7 / **91.9** / 80.8 / 90.8 | 79.5 / 93.9 / **98.4** / 96.0 / 98.2 |
| | CIFAR-10 | 19.1 / 18.7 / **20.2** / 12.2 / 16.3 | 77.1 / 77.2 / **77.5** / 67.9 / 75.3 |
| CIFAR-10 (ResNet) | iSUN | 44.6 / 73.2 / 97.8 / 99.3 / **100** | 91.0 / 94.0 / 99.5 / 99.8 / **100** |
| | LSUN(C) | 48.6 / 62.0 / 81.3 / 89.8 / **99.9** | 91.9 / 91.2 / 96.7 / 97.8 / **99.9** |
| | TinyImgNet(C) | 46.4 / 68.7 / 92.0 / 96.7 / **99.0** | 91.4 / 93.1 / 98.6 / 99.2 / **99.8** |
| | SVHN | 50.5 / 70.3 / 87.8 / 97.6 / **98.0** | 89.9 / 96.7 / 99.1 / 99.5 / **99.8** |
| | CIFAR-100 | 33.3 / 42.0 / 41.6 / 32.9 / **64.8** | 86.4 / 85.8 / 88.2 / 79.0 / **95.0** |
| CIFAR-10(DenseNet) | iSUN | 62.5 / 93.2 / 95.3 / 99.0/ **100** | 94.7 / 98.7 / 98.9 / 99.8 /**100** |
| | LSUN(R) | 51.8 / 70.6 / 48.2 / 88.4 / **90.1** | 92.9 / 93.6 / 80.2 / 97.5 / **98.0** |
| | TinyImgNet(R) | 56.7 / 87.0 / 84.2 / **96.7** / 83.2 | 93.8 / 97.6 / 95.3 / **99.3** / 95.1 |
| | SVHN | 40.2 / 86.2 / 90.8 / **96.1** / 90.3 | 89.9 / 95.5 / 98.1 / **99.1** / 98.2 |
| | CIFAR-100 | 40.3 / 53.1 / 14.5 / 26.7 / **58.3** | 89.3 / 90.2 / 58.5 / 72.0 / **92.3** |
| SVHN(ResNet) | iSUN | 77.1 / 79.1 / 99.7 / 99.4 / **100** | 92.2 / 91.4 / 99.8 / 99.8 / **100** |
| | LSUN(R) | 74.3 / 77.3 / 99.9 / 99.6 / **100** | 91.6 / 89.4 / 99.9 / 99.8 / **100** |
| | TinyImgNet(R) | 79.0 / 82.0 / **99.9** / 99.3 / 97.5 | 93.5 / 92.0 / **99.9** / 99.7 / 99.5 |
| | CIFAR-10 | 78.3 / 79.8 / **98.4** / 85.8 / 98.0 | 92.9 / 92.1 / 99.3 / 97.3 / **99.4** |
| SVHN(DenseNet) | iSUN | 78.3 / 82.2 / 99.9 / 99.4 / **100** | 94.4 / 94.7 / 99.9 / 99.8 / **100** |
| | LSUN(R) | 77.1 / 81.1 / 99.9 / 99.5 / **100** | 94.1 / 94.5 / 99.9 / 99.8 / **100** |
| | TinyImgNet(R) | 79.8 / 84.1 / **99.9** / 99.1 / 97.8 | 94.8 / 95.1 / **99.9** / 99.7 / 99.1 |
| | CIFAR-10 | 69.3 / 71.7 / **96.8** / 80.4 / 78.1 | 91.9 / 91.4 / **98.9** / 95.5 / 94.9 |

## 4.2 Ablation study

The OOD detection performance relies heavily on two key factors: the classifier's performance with in-distribution data and the quantity of training data accessible for density mapping. In contrast, the NF architecture has a relatively minor impact on OOD performance.

One can then safely say that the effectiveness of the proposed approach is primarily attributed to the capability of the classifier to push the ID logits far from the center, and the capacity of the NF technique accurately maps the density of the ID embeddings.

The impact of the classifier on OOD detection performance can be assessed by considering the flexibility of the classifier. One critical contributory factor to the flexibility of the classifier is the number of parameters and the architecture.

When conducting the genome experiment, an increase in complexity of the classifier model, characterized by a higher number of parameters, leads to a noticeable improvement in OOD detection performance (cf., table 3). While the improvement in validation accuracy for the classifier on ID data may not be significant as the classifier complexity increases, the embeddings produced by these classifiers exhibit a progressive separation from OOD examples (cf., table 3).

Table 3: Performance of OOD detection increases over the genome dataset over an increasingly more flexible classifier (increasing the number of parameters for Resnet-34).

| Nr Parameters (M) $\rightarrow$ | 3.7 | 3.9 | 5.3 | 5.5 |
|---|---|---|---|---|
| AUCROC $^{(\%)\uparrow}$ | | 75.9 | 75.8 | 82.1 | 86.0 |
| AUCPRC $^{(\%)\uparrow}$ | | 76.6 | 77.7 | 80.6 | 87.9 |

In order to assess the significance of NF performance as a density estimator, its architecture and the amount of training data are investigated. The success of the RealNVP architecture, which is the preferred choice for the NF model, is attributed to its affine coupling technique.

Table 4: OOD detection performance fashion-MNIST (ID) vs. MNIST (OOD) remains relatively stable while using different masks and repetition for each mask (in Appendix cf., figs. 23 to 25). The first column indicates the type of mask utilized. The second column indicates the metric, and the rest indicates the repetition number.

| Mask type and repetion. | | | | | |
|---|---|---|---|---|---|
| Mask type $\downarrow$ | Repetition $\rightarrow$ | 1 | 2 | 3 | 4 |
| Mask 1 | AUCROC $^{(\%)\uparrow}$ | 96.2 | 99.5 | 99.3 | 98.7 |
| | AUCPRC $^{(\%)\uparrow}$ | 97.0 | 99.5 | 99.5 | 99.1 |
| Mask 2 | AUCROC $^{(\%)\uparrow}$ | 98.8 | 99.2 | 99.2 | 99.2 |
| | AUCPRC $^{(\%)\uparrow}$ | 98.5 | 99.4 | 99.4 | 99.3 |
| Mask 3 | AUCROC $^{(\%)\uparrow}$ | 99.0 | 99.3 | 99.2 | 99.1 |
| | AUCPRC $^{(\%)\uparrow}$ | 99.3 | 99.5 | 99.4 | 99.3 |

Table 5: The amount of in-distribution data for density mapping remains a critical aspect of the OOD detection performance. The first column indicates the percentage of the ID data fashion-MNIST (ID) utilized for the training of the NF. The second and third columns indicate the model's performance at OOD (MNIST data) detection.

| Data size on training NF. | | |
|---|---|---|
| Data ratio | AUCPRC $^{(\%)\uparrow}$ | AUCPRC $^{(\%)\uparrow}$ |
| 0.1% | 71 | 75.5 |
| 1% | 92.9 | 94.2 |
| 10% | 98.2 | 98.7 |
| 25% | 98.5 | 98.9 |
| 50% | 99.1 | 99.4 |

An ablation is conducted to examine how different coupling of based distribution dimensions impact the OOD detection performance. Dimension coupling involves jointly transforming the coupled dimensions of

the based distribution (i.e., $P_X(x)$) instead of treating them independently. While maintaining the same model architecture for each dimension of the NF, three different masking methods (cf., figs. 23 to 25 in Appendix) that enable different types of coupling for the base distribution dimensions are tested (cf., table 4). In addition to masking, the number of repetitions of each MLP unit (cf., eq. (26) in Appendix ) is another crucial component of the NF. Hence, different masking operations are tested for up to four repetitions to investigate (cf., table 4).

Understanding the impact of training data size remains of highly practical relevance. Therefore, instead of training the NFs with all the training data utilized to train the classifier, a progressively smaller part is tried. The OOD detection performance is evaluated on the entire test data set (cf., table 5).

## 5  Discussion and Conclusion

Our work addresses OOD detection by building an accurate ID detection from logits as a one-class classifier and considering any non-ID as an OOD. This detection is possible as the method considers a maximal separation of OODs and IDs in the logit space. Under the assumption that their discriminative features originate from an altogether different distribution relative to the training data, it is possible to anticipate the interaction of OODs with the trained classifier parameters (cf. Section 2.2).

Whenever negative values are suppressed through the nonlinear function (i.e., ReLU), ID data are maximally distanced from the center towards the positive regions of the logit space (cf. Section 2.1). We show that the better the performance of the classifier on the ID data, the more distance their embeddings have from the center of the logit space, and the more compact their class-wise clusters are (cf., figs. 7 to 10 in Appendix). Another critically important aspect of an accurate classifier is that it embeds the OODs towards the center of the logit space such that they do not overlap with the IDs (cf. Section 2.2). The detection performance of OOD is driven by the scale of their statistical independence relative to the IDs (low covariability) and the accuracy of the classifier (high covariability between parameters and IDs). The more consistent this independence between OODs and IDs, the lower the expected magnitude of the OOD logits (cf., eq. (3)). Combining these two key drivers enables decoupling the IDs from the OODs in the logit space.

The proposed method demonstrates performance comparable to SOTA models on FPR at 95% TPR, AUCPRC, and AUCROC on both images and genome datasets. The primary factors driving the success of the proposed method are twofold: the accuracy of the classifier in the ID data and the thorough density mapping of the ID logit embeddings. Since ID data are sufficient for training classifiers and NF, the proposed method does not require exposure to simulated or gathered data intended to be OOD as in Tack et al. (2020); Winkens et al. (2020); Hendrycks et al. (2019). Using logit embeddings enables the formation of compact class-wise clusters. Utilizing a dedicated NF for each cluster in the logit space allows effective density mapping. By employing individual NFs for each designated class, the complexity of the architecture for each NF can be reduced significantly compared to using a single NF for all clusters combined.

This work has identified the importance of using ReLU as an activation function to separate OODs from IDs in the logit space and clustering the IDs compactedly towards the positive regions. However, one limitation of the current approach is that it requires a separate NF for every designated class, making it challenging to train setup when the number of classes is very high. Nonetheless, as part of our future work, we are actively exploring the development of a model that remains robust regardless of the number of classes.

Another promising future avenue is to leverage and replace the softmax with the NF setup (cf., fig. 1) for a better uncertainty estimation within the conformal prediction setup (Angelopoulos & Bates, 2022). Given that the proposed configuration of NF (cf., fig. 1) incorporates more effectively data uncertainty into the prediction response than softmax, one can use these NF scores to better calibrate the prediction via conformal prediction (Angelopoulos & Bates, 2022).

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
