# OpenReview forum: "Hiding in a Plain Sight: Out-of-Distribution Detection from Logit Space Embeddings"
_TMLR — Rejected by TMLR_

### Review · Reviewer_Qu5J · 2023-09-24

**Summary Of Contributions:**

This paper proposes a new method that detects out-of-distribution samples. Specifically, the authors propose using the normalizing flow to estimate the likelihood of the logit layer output for a given sample and a fixed classifier model.

The authors first provide an analysis that DNN classifiers with ReLU activation tend to map out-of-distribution samples near zero in the logit space. In summary, in logit space, the image of in-distribution samples forms clusters for each class, and each cluster is pushed outward from the center. On the contrary, the logit space image of out-of-distribution samples is concentrated near zero.

Then, based on the observation, the authors propose a new method for out-of-distribution sample detection. The idea is to measure the likelihood of the logit layer output being a logit output of an in-distribution class sample. The authors train a normalizing flow model for each class to measure this likelihood. If the logit layer output does not achieve significant likelihood for all classes, the given sample is classified as an out-of-distribution sample.

With experiments, the authors support the proposed detection method. To explain, as performance measures, the authors present the area under the ROC curve and the area under the PR curve. The performance of the proposed method is compared to many other existing methods, showing excellence in out-of-distribution sample detection. Then, ablation studies are presented to determine the factors that affect the performance of the proposed method.

**Audience:**

Yes

**Broader Impact Concerns:**

I don’t see a particular broader impact concern regarding this paper.

**Claims And Evidence:**

Yes

**Requested Changes:**

1. Regarding the organization of the paper, consider the following changes
  * Are the *consistently independent* random variables a common concept? If so, provide some reference, otherwise, briefly describe what consistently independent model parameters and out-of-distribution data mean in reality. Also, it looks like that in-distribution samples are consistently independent of the out-of-distribution samples. Explain why this is necessary from the definition.
  * If I understood correctly, the experiment has two main parts: performance evaluation (with a comparison to other models) and ablation study. Put a subsection for each and have a summary of key findings at the beginning of Section 4.
  * For each table, describe the key results in the caption or in the main text. It makes it easier to understand the key takeaways of the experiment section.
  * Some contents can be moved to the Appendix to add other details to the main body.
    * As long as the readers can understand how density estimation can be done by normalizing flow (specifically RealNVP), details about normalizing flow and RealNVP are not important enough to be in the main body of the paper.
    * The empirical validation in Section E of the Appendix is worth presenting in the main body to support the theoretical findings.
2. Consider addressing the following minor issues regarding the paper writing.
  * When citing other papers, distinguish in-text citations (`\citet`) and in-parenthesis citations(`\citep`). Please see the [Author guideline](https://jmlr.org/tmlr/author-guide.html) for the general rule.
  * Rather than having a single figure containing two subfigures, use `\subfigure` to split the plots and add captions to those subfigures to indicate what each figure represents. Adding an arrow and text between subfigures is possible to better render the “Affine transform” between two subfigures.
  * If the referred equation, figure, and table are in the Appendix, clearly say that they are in the Appendix. For example, figs. 11 to 13 (Page 9) do not appear in the main body of the paper. Also, does eq. (26) in Page 5 mean Equation 3? If not, eq. (26) is not in the main body, but it is an essential equation for discussing the paper, so move it to the main body of the paper.
3. The result in Table 5 can be emphasized further. Even with a data ratio of 1%, it shows performance better than other models. This result presents the data efficiency which is worth being presented.

**Strengths And Weaknesses:**

# Strengths
1. The analysis looks to be correct and contains a meaningful result.
2. The variety of the compared models and the datasets is impressive.
3. The experimental results seem promising.

# Weaknesses
1. The analysis uses a specific property of ReLU output (that all the outputs are non-negative) so it does not generalize to more general cases that use activations different than ReLU, e.g., sigmoid, leaky ReLU.
2. The proposed method might require normalized flow models depending on the number of classes. For example, CIFAR-100 contains 100 classes, and applying the proposed method will require training 100 normalized flow models for density estimation.
3. The authors should organize the paper in a better way. See the requested changes below.
4. There are several minor writing problems. See the requested changes below.

---

> ### Author Response · Authors · 2023-10-07
> **Comment about ReLU activation**
>
> Weakness 1:
>
> Dear Reviewer,
> We sincerely appreciate your valuable insight into this crucial key driver.
>
> Our empirical research demonstrates that when employing the ReLU, OOD data are best separated from ID data.
>
> Nevertheless, alternative activation functions that tighten the range of negative values, such as Leaky ReLU and others, also maintain this separation to some extent.
>
> We intend to incorporate comparative plots in the final experiments.

---

> ### Author Response · Authors · 2023-10-07
> **Comment about the number of normalizing flows**
>
> Weakness 2:
>
> Dear Reviewer, thank you for this important point.
>
> Indeed, the current approach faces challenges, mainly when dealing with a high number of classes, for two fundamental reasons.
>
> Firstly, as the number of classes increases, the requirement for a correspondingly high number of normalizing flows becomes apparent. This poses a significant computational burden and increases the complexity of the model.
>
> Secondly, the need for an extensive amount of training data for density estimation also escalates, as each of the normalizing flows must be meticulously trained.
> Failing to adequately train even a single normalizing flow could lead to inaccurate likelihood assignments for out-of-distribution data, ultimately undermining the overall model's performance.
>
> Nonetheless, as part of our future work, we are actively exploring the development of a model that remains robust regardless of the number of classes.
> The model presented in this work serves as the foundation for validating the effective separation of in-distribution (ID) and out-of-distribution (OOD) data within the logit space.

---

> ### Author Response · Authors · 2023-10-07
> **Comments about paper organsation**
>
> Weakness 2&3:
>
> Dear Reviewer, we thank you for your writing comments.
> We are working on all the suggested points, which will be included in the final version.

---

### Review · Reviewer_mYDk · 2023-09-25

**Summary Of Contributions:**

This paper makes the observation and theortical analysis on the distribution of ID and OOD samples in the logit space of neural networks. The ID samples tends to cluster towards the postive area far from the original. This enables the OOD detection with flow-based density estimator over training data logits, where an input in the low density area would be considered as OOD.

**Audience:**

Yes

**Broader Impact Concerns:**

No concerns on broader impact.

**Claims And Evidence:**

No

**Requested Changes:**

1. Clearly discussion the requirement on ReLU activation in the theortical derivations in Sec. 2.1
2. Given the discussion with logits and softmax, it would be interesting to have an ablation on the impact of different layer's output features on detecting OOD examples with the proposed method
3. Experiments should be conducted with more complicated dataset. Even CIFAR-100 experiments could help on the scalability concern.

Minor issue: The reference of equations in Sec. 2.2 is messy. Please fix.

**Strengths And Weaknesses:**

## Strength
1. This paper makes interesting theortical analysis on the distribution of ID logits in the logit space
2. The proposed method is straightforward and easy to follow.

## Weakness
1. This paper bases its discussion on setting where "ReLU is used as activation, so all logits are positive". However, in reality most model does not use ReLU at the last layer, so logits can also be negative. Though the theortical derivation may still hold with the output of the softmax (with is all positive), it would worth more clearification on how the logit is defined and whether the conclusion can be extended to other activations.
2. This paper designs the OOD detection method based on the theortical anlaysis on the distribution of logits in the logit space. However, the clustering of ID logits with minimal interclass distance and maximal intraclass distance may only be achieved in the optimal case where the data is fully seperatable and the model is fully optimized (0 training loss). This may not always be the case for deep leanring models on complicated training tasks, especially when only cross entropy loss is used without additional tricks. Previous research like [1] has shown that even for CIFAR-10 dataset specially designed objective is needed to make the logit distribution more clustered.
3. As mentioned in the related work section, previous work shows that softmax itself cannot capture sufficient uncertainty for OOD detection. Since logits have a direct mapping to softmax probabilities, it is unclear why logits can perform better than softmax for OOD
4. The scalability of the proposed method is questionable. As the proposed OOD detection method requires class-wise density estimation with flow-based models, more classes require more clusters. Furthermore the classes may not be fully separable with more classes. No large scale experiments are provided in the paper.

[1] https://openaccess.thecvf.com/content_cvpr_2018/papers/Wan_Rethinking_Feature_Distribution_CVPR_2018_paper.pdf

---

> ### Author Response · Authors · 2023-10-04
> **Comment about the different activation functions**
>
> Weakness 1: "How the logit is defined, and whether the conclusion can be extended to other activations"?
>
> Thank you for your time and your suggestion. We agree that aside from ReLU activations, various functions have been employed, and we have conducted an ablation study encompassing a range of different function types.
>
> Our findings remain consistent across various functions that limit or suppress the range of negative values, including:
>
> a) Leaky ReLU
> b) Celu
> c) Elu
> d) Selu
> e) Gelu
> f) Silu
> g) Mish
>
> These empirical validations will be incorporated into the final report, along with the corresponding experiment plots.

---

> ### Author Response · Authors · 2023-10-04
> **Comment about minimal interclass distance and maximal intraclass distance.**
>
> Weakness 2:
>
> We thank you for this valuable insight.
>
> As emphasized in [1], it is essential to note that the softmax output does not necessarily ensure the maximum separation in terms of distance between classes (minimizing interclass variance and maximizing interclass variance) in the logits space.
>
> Nevertheless, the softmax function does guarantee effective angular separation of both logits and softmax output.
>
> This maximal angular separation in the logit space is translated into a maximal distance separation in the logit space when negative values are suppressed and positive values are amplified.
>
> Specifically, achieving this angular separation relies on amplifying the correct logit cell toward the positive values while the rest of the logit values are pushed toward the proximity of zero.

---

> ### Author Response · Authors · 2023-10-04
> **Comment about the uncertainty carried by softmax**
>
> Weakness 3:
>
> Thank you for your time. We believe that softmax plays a crucial role in decreasing uncertainty when compared to the logit space.
>
> In the logit space, all values undergo exponentiation and are subsequently normalized within a range from 0 to 1.
>
> This exponentiation process amplifies larger values more than smaller ones.
>
> Following normalization, this operation causes smaller values to become even smaller and larger values to become even larger.
>
> As a result, softmax effectively reduces variance compared to the logit space, which can result in an insufficient representation of uncertainty.

---

> ### Author Response · Authors · 2023-10-04
> **Comment about the scalability**
>
> Weakness 4:
>
> Thank you for bringing this up, as it is a critical factor for the broader adaptation of classifiers.
>
> As you rightly pointed out, the current method requires as many normalizing flows as there are classes.
>
> However, class separation is done through both angular separation and Euclidean distance as long as the model is well-trained.
>
> Given that negative logit values are suppressed, the model is restricted to separate the classes towards positive values of the cell.
>
> This cell corresponds to the correct class index since the logits and softmax output have the exact dimensions.
>
> To validate the scalability, we are currently working on the CIFAR-100 to test the scalability of the method and will be including the results in the final report.

---

### Review · Reviewer_bJAa · 2023-09-26

**Summary Of Contributions:**

Authors propose a new OOD approach by estimating the density of data on the logit space.

**Audience:**

Yes

**Broader Impact Concerns:**

No concerns.

**Claims And Evidence:**

Yes

**Requested Changes:**

**Major comments**:

- My major request would be to add SOTA methods in OOD detection and not only the basic ones [1, 2, 3, 4, 5, 6]. It would strengthen your claim to also add different architectures, such as densenet or ViT.

- Also, mathematical notations are not properly introduced.

**Minor comments**:

- L6: "E.g., when classifying bacteria from genome sequences using a DL model, it is crucial to consider the presence of novel (i.e., OOD) bacteria"--> Add references

- Second Paragraph: add references to support your claims on DL and OOD.

- L2 of the second paragraph.--> A dot is missing before "However".

- General comments: Avoid re-defining abbreviation such as (DL) at the first line of the second paragraph or at the first line of section 2.1, it as already been defined first line.

- "To the best of our knowledge, this is the first work that identifies and analyzes this separation of OODs and IDs logits." I believe that this sentence is a bit overclaim. There are lot of work that developed OOD detector by looking into the logit induced space see e.g. [1,2].

- Theorem 1--> Magnitude values and notation $\hat{L}(j)$ are not introduced yet.

- "assymptotically"--> asymptotically

- "cf. Appendix appendices F to H)."

- "algorithm 1"-->  Algorithm 1; "appendix I" --> Appendix I. "section 2.3"--> Section 2.3. "corollary 1"--> Corollary 1.  (Have to be checked throughout in the paper.


- Avoid numbering all equations in the Appendix. The rule is to number only those that are referenced in the text.


**References**

[1] Weitang Liu, Xiaoyun Wang, John Owens, and Yixuan Li. Energy-based out-of-distribution detection. Advances in Neural Information Processing Systems, 2020.

[2] Rui Huang, Andrew Geng, and Yixuan Li. On the importance of gradients for detecting distributional shifts in the wild. ArXiv, abs/2110.00218, 2021.

[3] Dan Hendrycks and Kevin Gimpel. A baseline for detecting misclassified and out-of-distribution examples in neural networks. In International Conference on Learning Representations, 2017.

[4] Yiyou Sun, Chuan Guo, and Yixuan Li. React: Out-of-distribution detection with rectified activations. In NeurIPS, 2021.

[5] Yiyou Sun, Yifei Ming, Xiaojin Zhu, and Yixuan Li. Out-of-distribution detection with deep nearest neighbors. In ICML, 2022.

[6] Eduardo Dadalto Camara Gomes, Florence Alberge, Pierre Duhamel, and Pablo Piantanida. Igeood: An information geometry approach to out-of-distribution detection. In International Conference on Learning Representations, 2022.



[7] Dan Hendrycks, Steven Basart, Mantas Mazeika, Mohammadreza Mostajabi, Jacob Steinhardt, and Dawn Xiaodong Song. Scaling out-of-distribution detection for real-world settings. In International Conference on Machine Learning, 2022.

[8] Haoqi Wang, Zhizhong Li, Litong Feng, and Wayne Zhang. Vim: Out-of-distribution with virtual-logit matching. 2022 IEEE/CVF Conference on Computer Vision and Pattern Recognition (CVPR), pages 4911–4920, 2022.

**Strengths And Weaknesses:**

**Strengths**:

The Method is new and interesting. Empirical results seem to be promising.

**Weaknesses**:

The comparison of the proposed approach is weak mainly due to the lack of methods comparison. Tons of approaches have been introduced since ODIN and the Mahalanobis distance, and claiming that you perform well compare to the SOTA is totally overclaimed. Furthermore, the novelty of the proposed approach is limited.

Also, mathematical notations are not properly introduced. The claim of the equation (10) in the proof of the Corollary 10 seems coming from nowhere. And expectation higher than something, doesn't include that the empirical estimation of the quantity inside the expectation is also higher. For me, this proof is just false?

---

> ### Author Response · Authors · 2023-10-11
> **Comment about the mathematical notation**
>
> Comment: "Also, mathematical notations are not properly introduced. The claim of the equation (10) in the proof of the Corollary 10 seems coming from nowhere. And expectation higher than something, doesn't include that the empirical estimation of the quantity inside the expectation is also higher. For me, this proof is just false?"}
>
> Dear Reviewer, thank you for your time and depth of examination for our report.
>
> As you indicated, the step of going from analytical to empirical covariance estimation was not straight, and given time constraints, we could not find a mathematical rationale for this step.
>
> However, we revised the mathematical justification (updated in the new text) for the OOD placement using empirical covariance and not the analytical covariance instead.
>
> This part aims to build a mathematical intuition on why OODs are projected towards the center of the logit space by using empirical estimation of the variance as a surrogate measure for the statistical dependence and independence between the IDs model's parameters and OODs.

---

> ### Author Response · Authors · 2023-10-11
> **Comment about different architectures**
>
> Comment: " It would strengthen your claim to also add different architectures, such as densenet or ViT."
>
> We thank the Reviewer for this insightful suggestion.
> We decided to include Densenet in our experiments and experiment with ViT in future works.
> The experiments with Densenet are on the CIFAR-10 and SVHN datasets. The new results are on the updated report.
> Notice that the experiment on the grayscale images (Fashion MNIST vs MNIST) utilizes a small model different from Resnet, where all activations are replaced with ReLU.

---

> ### Author Response · Authors · 2023-10-11
> **Comment about the claims about the separation of the OODs and IDs in the logit space.**
>
> Comment: "To the best of our knowledge, this is the first work that identifies and analyzes this separation of OODs and IDs logits." I believe that this sentence is a bit overclaim. There are lot of work that developed OOD detector by looking into the logit induced space"
>
> We thank the reviewer for this recommendation.
>
> Indeed, many previous works identify the separation of the OODs from IDs using logit embeddings.
>
> However, we believe that our work is the first to show an empirical and analytical analysis of the OODs and IDs configuration in the logit space.
>
> Therefore, we updated this sentence in the main text to:
>
> "Although previous works have identified and experimented with the separation of OODs and IDs in the logit space, to the best of our knowledge, this is the first work that demonstrates the expected configuration of OODs and IDs logits."

---

> ### Author Response · Authors · 2023-10-12
> **Comment about additional SOTA method**
>
> Comment:"My major request would be to add SOTA methods in OOD detection and not only the basic one."
>
> We thank the reviewer for this suggestion.
>
> Through the presented work, we try to showcase that OOD detection is a joint effort of the classifier (Resnet, Densenet) that separates the ID from the OODs and the method utilized to quantify this separation (Multiple NF model).
>
> On the other hand, the recently proposed methods, however (to our understanding), try to attribute the OOD detection performance solely to the quantification of the OOD vs. ID separation.
>
> These recent methods assume that OODs are somehow separated from IDs.
>
> While we propose the conditions under which this separation is optimal.
>
> Thus, our words try to contribute towards a better understanding of OOD and ID separation and how to quantify this separation with comparable performance to the prominent baselines (reported in Table 2).
>
> In future work, we will further improve quantifying this separation relative to the recently proposed SOTA methods.

---

### Author Response · Authors · 2023-10-12
**Updated Report**

Dear Reviewers,

Thank you for your time and your individual feedback.

We addressed the majority of the requested changes by including the following:

1. Densenet classifier on the colored image dataset.

2. Experimenting with CIFAR-100

3. Experimenting with different activation functions

Furthermore, we addressed most of the minor comments suggested by all reviewers.

The updated main text and the Appendix have been uploaded.

---

### Decision · Action_Editor_JW3z · 2023-10-28

**Recommendation:** Reject

**Comment:**

This work contains an analysis about about the differences in the logit output distribution between in-distribution and out-of-distribution samples. Then, the paper discusses how to exploit the logit output distribution to detect out-of-distribution sample. In particular, the work proposes to use the normalizing flow as a density estimation algorithm to compute the probability density of the logit layer output for a given sample. If the logit layer output is unlikely to come out from in-distribution samples of any class, then the method conclude the sample as an out-of-distribution sample.

The paper provides well-supported claims and evidence, and I believe that the findings are nontrivial enough to intrigue other TMLR audiences. However, the current version still has some problems. For example,  (1) There are still mathematical details that are not rigorous. Furthermore, most of the OOD research now focus on new DNNs such as Vision transformers that will be more and more used in practice. (2) The analysis uses a specific property of ReLU output (that all the outputs are non-negative) so it does not generalize to more general cases that use activations different than ReLU, e.g., sigmoid, leaky ReLU. The proposed method might require normalized flow models depending on the number of classes. For example, CIFAR-100 contains 100 classes, and applying the proposed method will require training 100 normalized flow models for density estimation. (3) The improvement over baseline is not consistent, especially for hard detection problems like CIFAR-100 vs. CIFAR-10. Though this paper is based on interesting theortical analysis, the theortical derivation only works for an ideal case, which seems not always hold in practice for the problem scale of current deep learning research. The scalability issue of the proposed method is still questionable.

Therefore, we cannot accept this work this time, but the authors are encouraged to resubmit after a major and significant revision. We will consider to recommend its acceptance if the authors had addressed these issues properly.

**Audience:**

Yes

**Claims And Evidence:**

The current version still has some problems. For example,  (1) There are still mathematical details that are not rigorous. Furthermore, most of the OOD research now focus on new DNNs such as Vision transformers that will be more and more used in practice. (2) The analysis uses a specific property of ReLU output (that all the outputs are non-negative) so it does not generalize to more general cases that use activations different than ReLU, e.g., sigmoid, leaky ReLU. The proposed method might require normalized flow models depending on the number of classes. For example, CIFAR-100 contains 100 classes, and applying the proposed method will require training 100 normalized flow models for density estimation. (3) The improvement over baseline is not consistent, especially for hard detection problems like CIFAR-100 vs. CIFAR-10. Though this paper is based on interesting theortical analysis, the theortical derivation only works for an ideal case, which seems not always hold in practice for the problem scale of current deep learning research. The scalability issue of the proposed method is still questionable.

**Resubmission Of Major Revision:**

The authors may consider submitting a major revision at a later time.